# Grasping through dynamic weaving with entangled closed loops

Gyeongji Kang [1,2], Young-Joo Kim[3,7], Sung-Jin Lee [4], Se Kwon Kim [5], Dae-Young Lee [4,6] ✉ & Kahye Song [1] ✉

Pick-and-place is essential in diverse robotic applications for industries including manufacturing, and assembly. Soft grippers offer a cost-effective, and low-maintenance alternative for secure object grasping without complex sensing and control systems. However, their inherent softness normally limits payload capabilities and robustness to external disturbances, constraining their applications and hindering reliable performance. In this study, we propose a weaving-inspired grasping mechanism that substantially increases payload capacity while maintaining the use of soft and flexible materials. Drawing from weaving principles, we designed a flexible continuum structure featuring multiple closed-loop strips and employing a kirigami-inspired approach to enable the instantaneous and reversible creation of a woven configuration. The mechanical stability of the woven configuration offers exceptional loading capacity, while the softness of the gripper material ensures safe and adaptive interactions with objects. Experimental results show that the 130 g·f gripper can support up to 100 kg·f. Outperforming competitors in similar weight and softness domains, this breakthrough, enabled by the weaving principle, will broaden the scope of gripper applications to previously inaccessible or barely accessible fields, such as agriculture and logistics.

Pick-and-place is the primary act of robots spearheading the adoption of robotics technology across diverse industry sectors, including manufacturing, assembly, and logistics[1–3]. Robot grippers are essential elements in these applications, and a diverse array of gripper concepts has been proposed[4–6] from simple two-jaw grippers[7,8] to multi-finger grippers resembling human hands for complex forms[9,10]. These grippers boost efficiency and productivity through reliable operation in pick-and-place tasks.

As part of ongoing efforts to advance gripper technology, researchers have proposed a soft gripper concept, which employs the soft robotics principle that leverages the softness of materials[11–15]. In pick-and-place operations, ensuring the safe and stable handling of objects necessitates sophisticated sensing and control systems, often resulting in increased costs and maintenance challenges. Soft grippers present a compelling alternative by harnessing the inherent softness of their constituent materials, allowing the gripper structure to adapt to a shape of the object and achieve secure grasping without relying on complex sensing and control schemes[16–19].

Along with the softness of materials, various methods have been employed to achieve soft grasping action[20–25]. Multi-finger soft grippers resemble conventional grippers, employing a dynamic bending motion akin to human fingers, while the suppleness of the material enhances grip on the contours of the object[26,27]. Adhesion-based grippers, utilize structural design[28], electrostatic attraction[29], and

[1]Center for Intelligent and Interactive Robotics, Korea Institute of Science and Technology (KIST), Seoul 02792, Republic of Korea. [2]Department of Mechanical Engineering, Korea University, Seoul 02841, Republic of Korea. [3]Institute of Advanced Machines and Design, Seoul National University, Seoul 08826, Republic of Korea. [4]Department of Aerospace Engineering, Korea Advanced Institute of Science and Technology (KAIST), Daejeon 34141, Republic of Korea. [5]Department of Physics, Korea Advanced Institute of Science and Technology (KAIST), Daejeon 34141, Republic of Korea. [6]KAIST Institute for Robotics, Korea Advanced Institute of Science and Technology (KAIST), Daejeon 34141, Republic of Korea. [7]Present address: Center for Nanomedicine, Institute for Basic Science (IBS), Seoul 03722, Republic of Korea. ✉e-mail: ae_dylee@kaist.ac.kr; k.song@kist.re.kr

magnetic force[30], enabling soft grippers to cling to various surfaces, including smooth and rough ones[31]. Origami/Kirigami-inspired grippers, consisting of thin and flexible sheets of material, use three-dimensional structures to generate a particular grasping motion through mechanical inputs such as stretching or pulling[32–35].

While the inherent softness and adaptability of soft grippers offer substantial benefits, these characteristics simultaneously restrict their payload capabilities and render them vulnerable to external disturbances. As a result, these constraints narrow their scope of applications and hinder reliable performance, posing challenges to their integration into industrial environments. While there have been advancements in payload ratios[36–38], the absolute weight of the lifted object remains relatively modest, typically managing objects up to 100 mm in size and 3 kg·f in weight[39–42]. Standing out as outliers, Glick et al. introduced a soft gripper that utilized gecko-inspired adhesives, demonstrating a payload-to-weight ratio exceeding 150 for an object weighing approximately 11.3 kg·f[38]. Li et al. showcased a fluid-driven soft gripper inspired by origami, which lifted an object weighing 22 kg·f, and achieved a payload-to-weight ratio of 29[43].

In this study, we propose a weaving-inspired grasping mechanism that significantly enhances payload capacity while utilizing soft and flexible materials. By recomposing the principles of weaving, we engineered a flexible continuum structure capable of instantaneously forming and dismantling an enclosed space with high mechanical stability. Weaving typically involves constructing a continuous surface, such as textiles, from discrete elements like threads[44,45]. The mechanical entanglement of weft and warp threads yields stability, emulating the characteristics of a continuous substrate even though it is made up of individual segments[46,47]. Extensive research is being conducted on woven structures in diverse fields, including the medical field, owing to their remarkable homogeneity, stability, and capacity to interconnect complex structures without requiring specialized joining methods[48,49]. The proposed gripper design exploits these fundamental aspects of a woven structure, employing multiple closed-loop strips.

One main challenge of employing the weaving principle in the grasping mechanism is securing reversibility in constructing the woven configuration. To achieve this attribute, we have analyzed and simplified the conditions for weaving and used a kirigami-inspired approach to create a desired motion. Kirigami, a variation of origami, allows for greater flexibility in designing the motion of a continuum body through cuts in a thin substrate[50–52]. Based on this trait, the

proposed design allows rotational input to propagate throughout the entire structure, resulting in dynamic weaving motion, enabling the gripper to grasp objects. Because of the inherent mechanical stability induced by the given structure, it is possible to fully exploit the loading capacity of the material, and it can nearly perfectly prevent the failure of grasping within the range of the material's strength.

Experimental results show that the gripper, weighing 130 g·f, can hold up to 100 kg·f (~770 payload-to-weight ratio). Another notable feature of the weaving-inspired grasping mechanism is its softness and adaptability to objects. The structure is composed of thin, flexible materials (0.4 mm thick polyethylene terephthalate, PET), and each part is loosely entangled to allow movement relative to one another, allowing it to safely interact with objects even though it offers >100 kg loading capacity. To the best of our knowledge, the performance of the gripper outperforms its competitors in a similar domain of weight and softness, and this breakthrough is made possible by the principle of weaving that has not been used before. We believe the gripper can be widely adapted to fields that are currently inaccessible or barely accessible, such as agriculture and logistics.

## Results

A woven structure maintains structural stability through the intertwined configuration and tight entanglement of the warp and weft threads, which prevent it from being broken easily (Fig. 1a, b). The structure features threads that exist independently, are interlaced, and support each other by moving linearly to the center with position differences (the difference in the position of each thread is described by the (+) and (−) signs. For example, the white one is lower (−) than the red one in the z-direction, but higher (+) than the blue one in the z-direction) (Fig. 1c). Two notable features exist in the woven structure: (1) the lines are collected to the center by moving through linear motion, and (2) the lines have position differences along the z-axis relative to the adjacent lines. We focused on these two principles to design a weaving gripper that operates by intentionally switching between woven and unwoven states. The weaving gripper implements these principles by converting relative rotational motion into linear motion. In Fig. 1c, the horizontal direction (parallel to the y-axis) represents the weft, indicated by the presence of red and blue strips and the vertical direction (parallel to the x-axis) corresponds to the warp, denoted by the white and yellow strips (set based on the orientation of the current figure). By connecting the ends of each strip

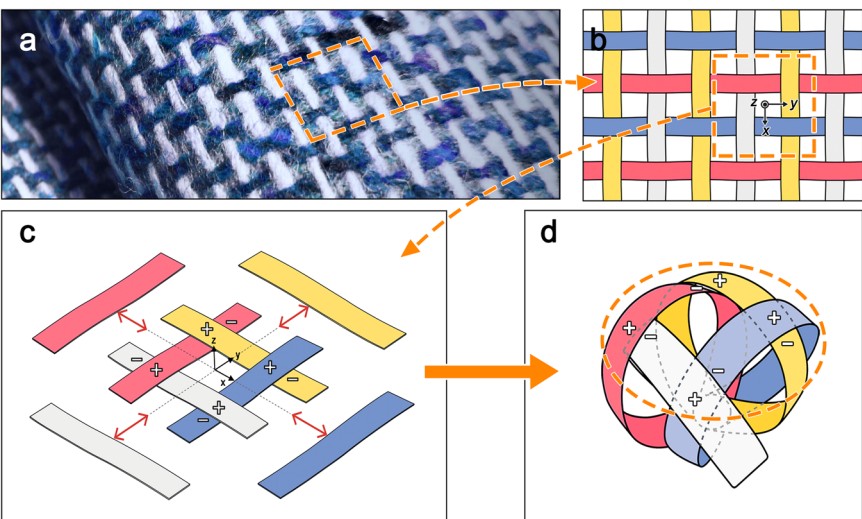

**Fig. 1 | Principle of the woven structure. a** Woven structure in fabric. **b** Threads form the warp and weft with position differences along the z-axis. **c** Every thread intersects at the center with a position difference in the z-axis by moving linearly.

Adjacent threads are each at a relatively high (+) or low (−) position in the z-direction. **d** Weaving mechanism reconfiguration for gripper by connecting the end of each strip with making a closed loop.

in sequence, respectively, the lines of the strips can be changed into multiple closed loops that intertwine with each other (Fig. 1d). The weaving mechanism then can be adapted for the gripper and exhibits significantly enhanced compositional strength as the weaving is completed with the gripper in the closed state. The strips can be brought together linearly towards the center, resulting in a woven state, while relative rotation moves the strips to an unwoven open state. It remains in an open state before grasping, where the lines do not intersect. When relative rotation is applied, the lines move linearly toward the center, forming a spherical shape, and finally becoming woven.

The gripper consists of multiple identical closed-loop strips, each serving as both a warp and a weft. Each strip was designed as a single line connecting two circular plates (outer and inner plate) with different sizes at each end (Fig. 2a). The middle part of the strip is twisted by 180° to form a loop so that the upper faces of the plates at both ends of the strip face are equally upward. Inner and outer bound points exist, which are the boundaries of the plates and strip (Fig. 2b). The boundary between the inner plate and strip is the inner bound point, and the boundary between the outer plate and strip is the outer bound point. Owing to the input rotation value $\theta_i$ of the inner plate, an angular difference occurs between the tangents at the inner and outer points, which affects the shape of the loop and ultimately enables the weave. Additionally, two offsets exist between these two points (Fig. 2b, c): the differences in the plate size and thickness of the stacked strips cause radial position and height offsets denoted by $r_{offset}$ and $z_{offset}$, respectively. In the weaving process, the aforementioned offsets enable the transition between the woven and unwoven states without interference between the strips. The outer plate at the bottom is fixed to the frame, and the inner plate at the top is connected to the electric motor. By rotating the inside plate clockwise, we induced strip motion by creating a relative rotational motion between the two plates (Supplementary Video 1). The strips were placed circularly along the inner and outer ground. To explain the weaving mechanism, we simulated the structural changes of the weaving gripper with eight strips through finite element analysis (FEA) and analyzed its trajectories at different rotation angles. For clarity, only the reference strip and the preceding and following strips in the clockwise position changes are illustrated in Figs. 2f, i, l, o, and r.

At $\theta_i = 0°$, the $\theta$-axis values of the inner and outer bound points match and the gripper is in an unwoven state (Outer bound points and inner bound points are on the outer ground line and inner ground line, respectively) (Fig. 2d–f). The independent strips do not cross each other because they are sequentially arranged with original structural position differences in the $x$-$y$ plane and along the $z$-axis. When the inner plate is rotated clockwise such that $\theta_i = -22.5°$, the inner ground point shifts to −22.5° (Fig. 2g–i). When $\theta_i = -45°$, the $\theta$ values of the outer bound point of the preceding strip and the inner bound point of the reference strip coincide by rotating clockwise (Fig. 2j–l). However, owing to the differences in the original position difference in the $x$-$y$ plane and along the $z$-axis, the strips do not collide and always overlap with respect to the $z$-axis with a certain rule: the reference strip is always located lower than the preceding strip clockwise (Fig. 2l **A**) and higher than the following strip on the $z$-axis (Fig. 2l **B**), in the exact same way that warp and weft threads are interlaced in the woven structure (Fig. 1c); when $\theta_i = -90°$, the inner bound point of the reference strip overtakes the outer bound point of the preceding strip along the $\theta$-axis (Fig. 2m–o). Then, an intersection between the strips occurs (Fig. 2o **A** and **B**), whereas the reference strip is still positioned lower along the $z$-axis than the preceding strip and higher than the following strip. $\theta_i = -180°$ is the angle $\theta_s$ at which the inner and outer boundary points within each strip are symmetrical (Fig. 2p–r). At $\theta_s$, the strips pass near the center point, and the number of overlapping intersections between the strips increases. Eventually, the intersection points of all the strips are firmly formed at positions lower than those

of the preceding strips and higher than those of the following strips along the $z$-axis (Fig. 2r **A** and **B**), forming the most stable woven state.

To use weaving in a gripper, it is necessary to be able to hold or release an object by using the relative rotation of the strips. The gripper performance was determined by the completeness of the weaving formed by the warp and weft strips surrounding the object. The completeness of weaving was quantified through an imaginary inscribed circle with radius $R_{incircle}$, where the inner edges of the strips were tangents (Fig. 2b and Supplementary Fig. 1). As the input angle increases, the interlacing of the strips becomes denser and the size of the incircle decreases. If there is no reaction force from an object, $R_{incircle}$ approaches 0 as $\theta_i$ approaches −180°, and it becomes the closest to the woven state. This state was set as the critical point at which the gripper reached the weave. To maximize the load capacity of the gripper, it is crucial to approach the critical point where there is a reaction force, and the gripper can be reached through a complementary relationship between $\theta_i$ and $R_{incircle}$.

If an object exists inside the gripper, a reaction force is generated by contact between the gripper and the grasping object. To understand their interactions during grasping, we measured the (in-plane) reaction force when a strip was rotated by $\theta_i$ and then pushed outward to $R_{incircle}$ through FEA, and the results are shown in Fig. 3a. When the input angle is small and $R_{incircle}$ is large, no interaction occurs between the object and gripper, so the gripper can "freely deform" (orange arrow in Fig. 3a–d). However, as the input angle increases and $R_{incircle}$ decreases, the gripper starts contacting the object. The gripper strip "adaptively deforms" (red arrow in Fig. 3a–d) along the object boundary, requiring increasingly larger reaction forces.

Assuming that the strips form an ideal sphere at $\theta_s$ ($\theta_i = -180°$), the sphere diameter is approximately 90 mm (Supplementary Fig. 2). All grasping experiments were performed on three balls of different sizes: a small 75-mm-diameter ball ($B_{75}$, 80% size, Fig. 3b); a 90-mm-diameter ball ($B_{90}$, 100% size, Fig. 3c), the same size as the ideal sphere; and a large 105-mm-diameter ball ($B_{105}$, 120% size, Fig. 3d). In the experiment, we provided an additional rotation angle $\theta_a$ ($\theta_a = |\theta_i - \theta_s|$) from the woven state, considering the volumetric effects of an object. Depending on the object size, $\theta_a$ at which the strips contacted the surface and reached the woven state differed. The larger object had limitations in reaching the same $R_{incircle}$ and could reach the same state by increasing $\theta_a$. An object of 80% size compared to the internal gripper volume reached the woven state at $\theta_s$. If we gave more angles, the strips became mechanically entangled while increasing adaptability to the ball. In the 100% size case, the gripper started to contact the ball at $\theta_i = -160°$ because of the eccentricity of the gripper and woven at $\theta_i = -210°$. The contact angle was different for the 80% size ball, but it could reach the same state by increasing the rotation by approximately 30°. On the other hand, the 120% object size exceeded the limit that could be supplemented with an angle.

To evaluate the structural performance of the dynamic soft weaving gripper experimentally, the load capacity and torque were measured according to the number of strips $N$ and rotation angle $\theta_a$ (Fig. 4a, b). We tested three grippers with an outer ground radius $R_o$ of 35 mm, but different $N$ (8 ($N_8$), 12 ($N_{12}$), and 16 ($N_{16}$)) (Supplementary Figs. 3 and 4), whose widths were determined to ensure that the gripper strip surface area was the same ($N \times$ strip width ($W$) $\cong 2\pi R_o$). Structural failure was defined as a case in which the gripper missed an object because its grasping force was less than the object's weight. However, according to the literature, we set a load of 100 kg·f or more as maximum load measurement limit $L_{max}$ to distinguish material failure from structural failure (Supplementary Fig. 5)[33].

When $\theta_a = 1°$ in the $B_{75}$ experiment, the loading capacity exceeded 100 kg·f and reached a $L_{max}$ (Supplementary Fig. 6). Regardless of $N$, all grippers could lift objects over 770 times their weight (130 g·f).

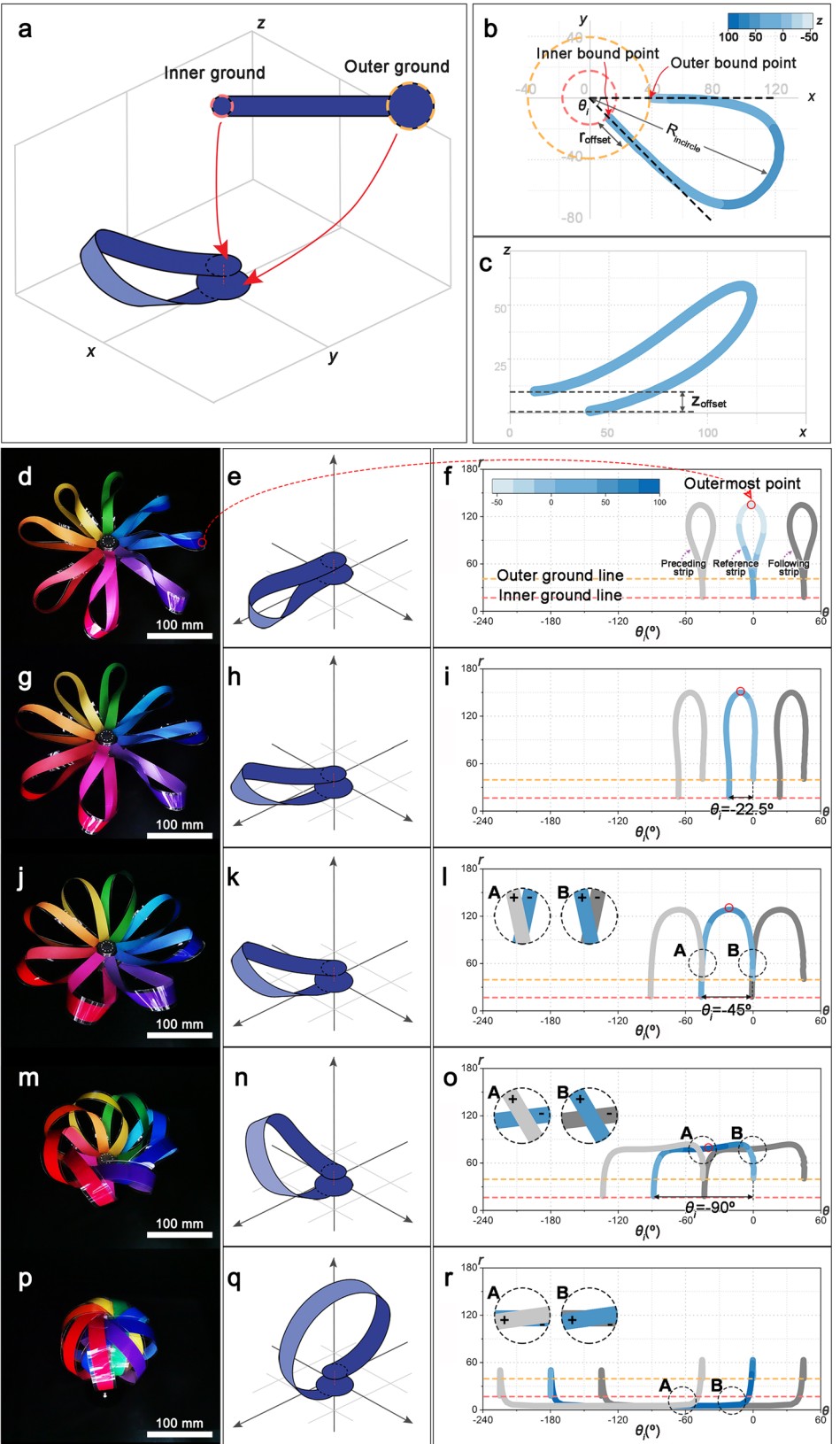

**Fig. 2 | Weaving gripper mechanism. a** Weaving gripper composed of strips, where one strip is designed as a line shape with inner and outer ground plates. **b** Schematic of one strip forming a loop on the $x$–$y$ plane (color bar representing the range of $z$ coordinates) and (**c**) on the $x$-$z$ plane showing $z_{offset}$ by stacked strips. **d**–**f** State of the weaving gripper and one strip showing the process of intersection between adjacent strips based on the gripper with eight strips at $\theta_i = 0°$, **g**–**i** $\theta_i = -22.5°$, **j**–**l** $\theta_i = -45°$, **m**–**o** $\theta_i = -90°$, and **p**–**r** $\theta_i = -180°$ (color bar representing the range of $z$ coordinates).

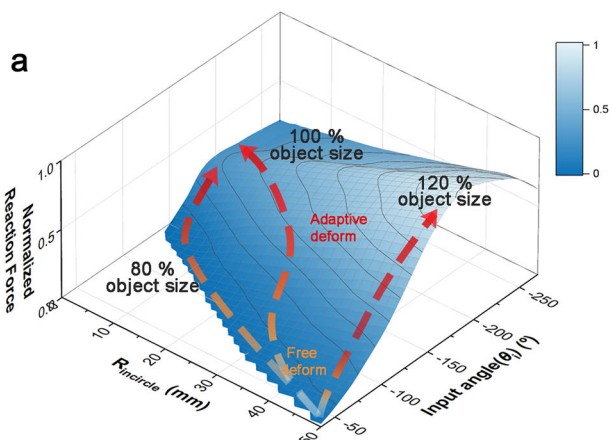

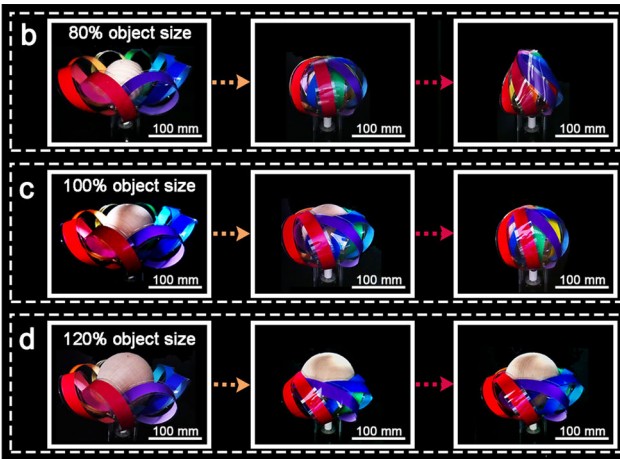

**Fig. 3 | Force prediction and deformation analysis depending on object size.** **a** Reaction force of the gripper prediction using FEM according to $R_{incircle}$ and $\theta_i$. The reaction force changes according to the degree of deformation of the gripper and the state of contact with the grasping target. **b** Weaving process with an 80%

object size compared to the internal gripper volume. $N_8$ contacts the surface of the object at $\theta_i = -180°$ and twists at $\theta_i = -240°$. **c** The 100% object size contacts at $\theta_i = -160°$ and is woven at $\theta_i = -210°$. **d** The 120% object size contacts at $\theta_i = -135°$ and still is not woven at $\theta_i = -190°$.

In contrast, in the $B_{105}$ experiment that did not reach the weaving state, the average maximum loading weight $W_{max}$ was less than 2 kg·f regardless of $\theta_a$. Rather, as the rotation angle increased, the strip was twisted, and the internal gripper volume decreased; therefore, $W_{max}$ was smaller. Additional tests were conducted using an ellipsoid ball having the same volume as the $B_{90}$ and an eccentricity of 0.9 (major axis $x = 156.6$ mm, minor axis $y$, $z = 68.2$ mm). As the height of the ellipsoid fully entered the interior of the gripper, the woven structure was completed at $\theta_a = 1°$, reaching an $L_{max}$. In contrast, balls larger than the gripper caused structural failure for sufficiently heavy loads. Balls larger or smaller than the gripper showed non-discriminatory results regardless of $\theta_a$ and $N$; therefore, the difference in performance according to $\theta_a$ and $N$ was compared using a 90 mm ball assumed to be of similar size to the internal gripper volume.

The performance of the gripper was found to depend on $N$ (Fig. 4c). The $N_{16}$ had the most strips, and thus the thinnest width, it was relatively flexible compared with $N_8$, and deformation occurred easily. However, $W_{max}$ was relatively low owing to the weak rigidity of the strip. $W_{max}$ measured on average at $\theta_a = 1°$ was approximately 2.78 kg·f with $N_{16}$ (The blue boxplot in Fig. 4c shows the results of five experiments with $N_{16}$, and the small square in the center of the boxplot shows the average value of $W_{max}$), and 4.04 kg·f with $N_8$ (The red boxplot in Fig. 4c shows the results of five experiments with $N_8$), approximately 1.5 times the value for $N_{16}$. The performance difference between samples with different $N$ values became clearer as the rotation angle increased. It was approximately 6.8 times at $\theta_a = 20°$, measured at approximately 6.15 kg·f with $N_{16}$ and 41.60 kg·f with $N_8$. The results show that a gripper with fewer strips reaches a critical point at a smaller $\theta_a$.

In conclusion, the control of the rotation angle $\theta_i$ between the two plates is a key factor in weaving. The load capacity increases exponentially beyond a certain critical angle, which depends on the number of strips. This tendency occurs because, as the rotation angle increases, the required force for the object to pass through the center point of the strip increases, and this force is directly related to the success of weaving. At $\theta_a = 1°$ and $4.5°$, the payload $W_{max}$ was measured to be 6.5 kg·f or less; the woven structure was not reached before $4.5°$. In comparison, it showed more than two times the performance at $\theta_a = 20°$ and ten times the performance at $90°$ compared to $4.5°$ (Supplementary Fig. 7). When $\theta_a = 0°$, the 90 mm ball was averagely wrapped only up to 88 mm, which was a height of 98% from the top of the ball, so the weaving was not formed yet. Rotating by more than $20°$

brought the distance between loops closer to zero sizes of $R_{incircle}$ and improved the performance by more than 10 times. Consequently, to maximize the loading capacity of the gripper, it is essential to make $R_{incircle}$ as small as possible close to the point or, equivalently, to make the interlacing of the strips as dense as possible.

The gripper exhibited a high mechanical efficiency compared to the generated torque (Fig. 4d), which was especially marked by the size of the object. The smaller objects $B_{75}$, which was fully gripped and reached the critical point at $\theta_s$, $W_{max}$ was measured over an $L_{max}$ with a torque of less than 2.5 N·m. In contrast, the larger objects $B_{105}$, which failed to reach the woven state and experienced slipping, demonstrated relatively low torque and payload. A load of approximately 2% was measured with $B_{105}$, whereas the generated torque was similar to that of $B_{75}$.

### Demonstration
The proposed principle can be applied to understand the mechanisms, irrespective of their scale. This claim was verified by the experiment of a large version ($G_l$) four times the scale and a small version ($G_s$) 1/5 the scale of the original model ($G_m$) (Fig. 5a). $G_l$ lifted a 5 kg·f box and 6 packs of 2 L bottled water (12 kg·f) (Fig. 5b, c). $G_m$ grasped objects such as a thin card (thickness: ~0.15 mm), four golf balls (diameter: ~43 mm), three of 30 kg·f dumbbells (90 kg·f), and a fragile flower (Fig. 5d–g). $G_s$ lifted a 500 g·f weight, which was more than 290 times the weight of the main body (1.7 g·f), and a coin (Fig. 5h, i). All related videos can be found in Supplementary Video 2–4 (Video of load capacity evaluation, adaptability evaluation, gripper size variation is shown in Supplementary Video 2, 3, and 4, respectively).

### Discussion
The primary focus of the present study was to explore the potential of a weaving-inspired grasping mechanism capable of substantially enhancing payload capacity while employing soft and flexible materials. The capability to lift heavy objects and secure stable grasping are the primary challenges that persist within the realm of soft grippers. Soft grippers typically weigh between 1 g·f and 1 kg·f with a payload of up to 3 kg·f, while conventional grippers typically weigh between 1 kg·f and 100 kg·f with a payload of more than 2 kg·f (See Supplementary Fig. 8 for the comparison chart of our gripper with other soft grippers and conventional grippers in terms of payload and gripper weight[53]). Soft grippers exhibit an advantage concerning weight, and conventional grippers show superiority in payload capacity, but the strength

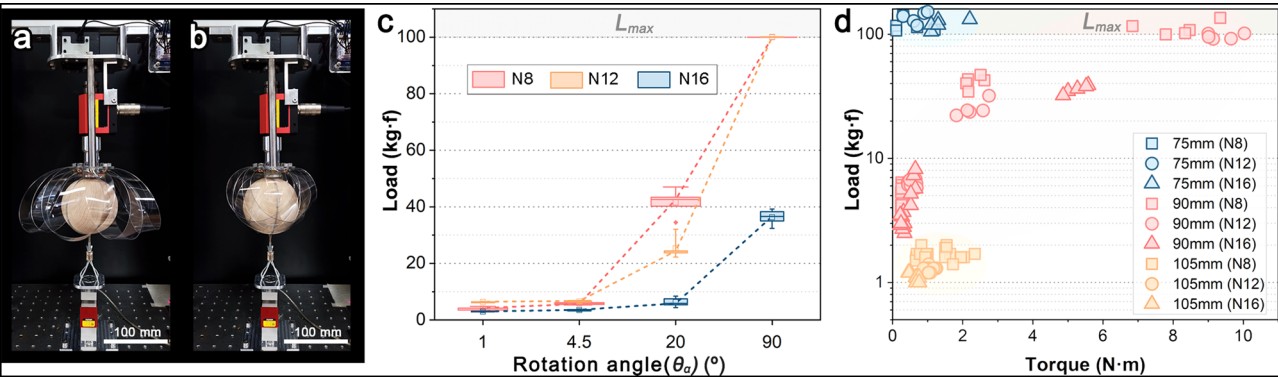

**Fig. 4 | Experimental analysis for weaving gripper performance.**
**a**−**b** Experimental setup for measuring load capacity and torque. **c** Load capacity measured depending on $\theta_a$ (red, yellow, and blue correspond to $N_8$, $N_{12}$, and $N_{16}$, respectively). **d** Load capacity results for torque according to object size (blue, red, and yellow correspond to $B_{75}$, $B_{90}$, and $B_{105}$, respectively).

of each gripper type inherently corresponds to the other's weakness. Whereas, the weaving gripper that we demonstrated in this study shows lightweight (130 g·f) and high payload (≥100 kg·f) with a weight-to-payload of 770. The weaving structure is known for its strong resilience to shape deformation (induced by, e.g., the payload on it) due to the cooperative efforts to sustain its shape by the constituent strips. This strong geometric resilience of the weaving structure employed in our gripper allows the individual strips to handle heavy payloads by distributing and supporting the weight. Additionally, the reversible structure formation speed of the gripper is directly influenced by the rotational speed of an electrical motor, facilitating rapid operation depending on the motor performance.

The flexibility of the material enables to grasp a wide range of objects. In our study, our primary focus was on examining spherical-shaped objects that resemble the inside of the gripper. We emphasized the variation in the number of strips based on the object's volume and explored how this variation affects performance. As the gripper is designed with a spatial volume resembling a sphere, objects with a size equal to or less than the gripper's internal volume can be securely gripped even if they possess low hardness or fragility. Regarding the shape of the object, it is primarily irrelevant to the grasping performance. However, if the volume of the object exceeds the volume of the gripper, the presence of an additional structure capable of forming a weave structure (such as a dumbbell-like structure) can be advantageous for grasping a high weight of tens of kilos or more. The gripper is also capable of securely grasping thin and flat objects, such as a thin card, utilizing its thin beveled edges and exerting force trying to reach $\theta_i = 0$ (the woven state).

Continued efforts in advancing both material and structure design aspects are essential to broadening the applicability of grippers across various fields. The gripper showed a high payload capacity, yet there remains significant room for further improvement. Once the woven structure is completely constructed, it remains intact until material failure occurs. This suggests that by employing materials with higher tensile strength, such as carbon-fiber composites, the payload capacity could be further dramatically enhanced. Additionally, changing the material can result in different physical properties of the gripper, such as grasping force or adaptability[54].

Moreover, the extensive design flexibility of the gripper enables it to meet a wide range of application-specific requirements, including object size, grasping force, adaptability to object shape, and robustness against external disturbances. As demonstrated by the divergent outcomes emanating from changes in strip count, the performance of the weaving gripper is predicated on its parameters, engendering varying results contingent on their structural configurations. The parameters impacting performance encompass

multiple factors, such as strip width, thickness, plate size, and structural patterns. Additionally, while the present study employed a straight-line strip, the strip can be designed to a variety of shapes, with the potential for enhanced functionality depending on the design. Consequently, a comprehensive evaluation of the design parameters is imperative to customize the gripper to particular applications to ensure optimal performance and superior outcomes in particular applications.

## Methods
### Materials and fabrication method
The system consists of a gripper, a motor, and a rotary shaft. The gripper is manufactured by stacking multiple identical strips consisting of a PET (Polyethylene terephthalate) film. Different thicknesses of the film are used for gripper models based on the size of grippers: 250 μm for $G_s$, 500 μm for $G_m$, 1.5 mm for $G_l$. One strip of $G_m$ has a total length of 400 mm and plate diameters of $\phi70$, and $\phi30$ (Supplementary Fig. 3). According to the number of strips, the polygon inscribed in the large plate is determined, and accordingly, the width of the strip and the diameter of the small plate are determined: widths were determined to ensure that the gripper strip surface area was the same ($N \times$ strip width ($W$) $\cong 2\pi R_o$). The smaller version $G_s$, reduced scale of 1/5, features plate diameters of $\phi14$ and $\phi6$ with a 3 mm wide strip, resulting in an unfolded diameter of approximately 55 mm. And the larger version $G_l$, enlarged to 4 times the scale, has plate sizes of $\phi280$, and $\phi80$, a strip width of 75 mm, and an unfolded diameter of approximately 1.2 m.

The grippers are fabricated by cutting strips with a $CO_2$ laser and assembling them by stacking the outer plates clockwise at a specified angle, then securing them with bolts. The inner plates are fastened with the same orientation as the outer plates, facing the same direction. The system is completed by connecting the inner plates to the rotary shaft and motor and fixing the outer plates to the frame. When the motor rotates clockwise, the strips interweave at the center to grasp the object.

### Experimental method
The motor from Robotis (XM540-W270-R, Robotis) was used for rotation, and manufactured the rotary shaft by processing AL6061.

The experiments were repeated five times for consistent parameters. In each trial, the gripper was moved from the initial angle $\theta_i = -70°$ to final angles $\theta_a = 1°, 4.5°, 20°, 90°$ and, then subjected to a vertical pull until failure occurred. The maximum load was measured by connecting wooden balls to the load cell via a steel wire and pulling the ball at a PWM frequency of 8 kHz using a 56-angle stepper motor (A16K-G268, Autonics) and two z-axis stages (LS1002-220-T56.4,

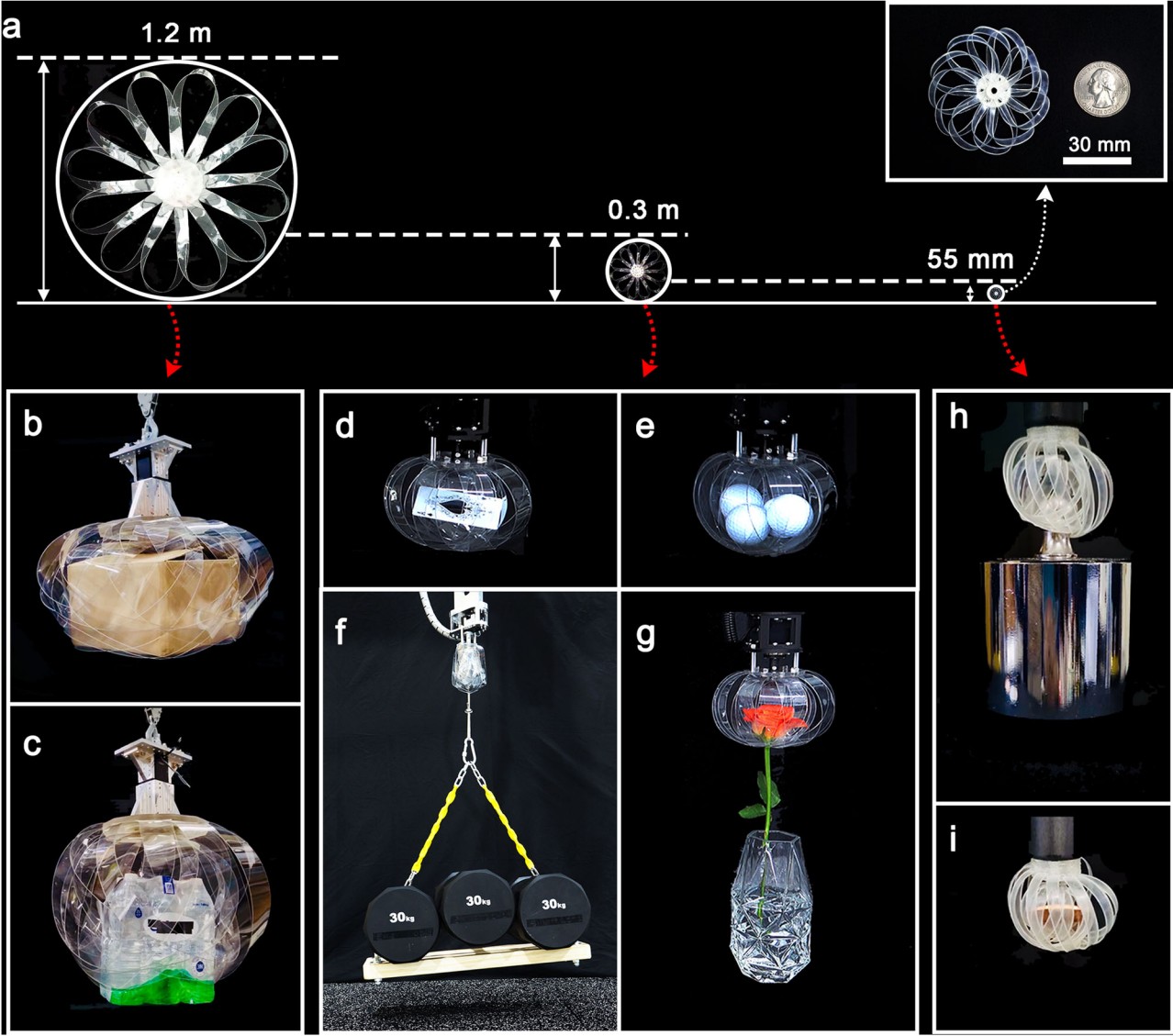

**Fig. 5 | Application experiment with three scales of weaving grippers. a** Weaving grippers manufactured on three scales: large scale ($G_l$) with a diameter of 1.2 m, medium scale ($G_m$) with a diameter of 0.3 m, and small scale ($G_s$) with a diameter of 55 mm. $G_l$ lifted (**b**) a 5 kg·f box and (**c**) 6 packs of 2 L bottled water (12 kg·f). $G_m$ lifted (**d**) a thin card, (**e**) four golf balls, (**f**) three 30 kg·f dumbbells (90 kg·f), and (**g**) a flower. $G_s$ lifted (**h**) a 0.5 kg·f weight, and (**i**) a 1 cent coin.

Misumi). A torque sensor (TRD605-160N.m, Futek) was installed between the motor and rotary shaft of the actuator to measure the torque generated during gripper operation. And a pointer laser sensor (IL-600, Keyence) was positioned on the z-axis stage to measure the height to which the ball was vertically moved. The maximum load capacity was measured using a load cell (LSB350-500lb, Futek). The data was collected in real-time at a rate of 400 Hz through the use of LabVIEW and a DAQ (NI USB-6343, National Instruments). And the motor was programmed to shut off automatically when the load exceeded a $L_{max}$.

### Application experimental method
Dynamixel motors from Robotis (Dynamixel XM540-W270-R, Dynamixel PH54-200-S500-R, Robotis) was used for the $G_s$ and the $G_l$, respectively. The rotary shaft and frame for the $G_s$ were printed using a 3D printer (Onyx Pro, Markforged). In $G_m$ performance test, $G_m$ was mounted on the end effector of the Robot Arm (A0912, Doosan Robotics). In the $G_l$ test, an aluminum frame of 1.5 m × 1.5 m × 2 m was set up, and a hoist was attached to the top to fasten $G_l$, allowing for its vertical movement.

### Numerical simulation
Essentially, the deformations that occurred in this study involve large displacement and strain in thin shell structures with both bending and twisting, making the structural morphing complex. Due to the difficulty of predicting them with a simple theoretical model, numerical simulations were employed to understand the complex morphing and mechanical behavior of the grippers in detail. Using a commercial finite element analysis (FEA) software package, ABAQUS (Standard), the PET strips were modeled as linear elastic, isotropic material with Young's modulus (E) of 3.5 GPa and Poisson's ratio (ν) of 0.38. Here, we did not consider the plasticity of the PET film to simplify the system without the loss of generic features in deformation. The geometries were meshed with a solid quadratic brick element (C3D20R). Since large displacement was involved in the grasping procedure, geometrical non-linearity was considered for all simulations. In Fig. 3, we performed numerical experiments for a single strip to get some insights during the grasping process. First, we bent a straight strip to form a purely bent shape ($\theta_i = -180°$). Then, we measured the in-plane reaction forces while rotating and pushing it outward from the center point. Specifically, we rotated the strip to $\theta_i$ and then applied

the in-plane displacement without constraining its vertical movement, pushing the middle part outward so that the center inner face of the strip is on $R_{incircle}$.

## Data availability
All data generated or analysed during this study are included in this published article (and its supplementary information files).

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

## Acknowledgements

We are grateful to Dr. D.-H. Kim (KIST) for discussion, to D. Choi (SWU) for schematic drawings, and to C. Jeong (POSTECH) for data acquisition.; This work has been funded by: Korea Institute of Science and Technology Institutional Program grant 2E32304 (G.K, K.S); Basic Science Research Program through the National Research Foundation of Korea funded by the Ministry of Education grant NRF-2020R1A6A3A01099512 (Y.-J.K); BK21 FOUR Program of the National Research Foundation Korea(NRF) grant funded by the Ministry of Education(MOE) (D.Y.-L) and National Research Foundation of Korea funded by the Ministry of Science and ICT grant 2022R1C1C1003718 (S.-J.L, D.-Y.L); Brain Pool Plus Program through the National Research Foundation of Korea funded by the Ministry of Science and ICT grant NRF-2020H1D3A2A03099291 (S.K.K).

## Author contributions

G.K, D.-Y.L and K.S conceived the initial concept and methodology, analyzed the experimental results, and contributed to writing the original manuscript. G.K, S.-J.L, D.-Y.L and K.S conducted experiments. Y.-J.K, S.K.K, D.-Y.L and K.S. contributed to theoretical analysis. G.K, Y.-J.K, S.K.K, D.-Y.L and K.S reviewed and edited the final manuscript.

## Competing interests

The Authors declare the following competing interests, 1. Patent application in Korea—Patent applicant: KIST & KAIST—Application number: 10-2-22-0012547—Name of inventors: Kahye Song, Gyeongji Kang, Dae-Young Lee—Status of application: Approval of patent registration—Specific aspect of manuscript covered in patent application: Fabrication method and configuration. 2. PCT Patent application—Patent applicant: KAIST—Application number: PCT/KR2023/001252—Name of inventors: Kahye Song, Gyeongji Kang, Dae-Young Lee—Status of application: Pending—Specific aspect of manuscript covered in patent application: Fabrication method and configuration. The Authors declare no other competing interests.
