## [Peer Review file · Nature Communications]

REVIEWER COMMENTS

Reviewer #1 (Remarks to the Author):

In the paper titled as “Grasping through dynamic weaving with entangled closed loops”, the authors provide a novel technique by using interwoven structure as gripper with high payload-to-weight ratio. The authors studied the grasping mechanism with sufficient details. Gripper with grasping mechanism initiated by the weaving of closed-loop strips is novel and promising. Before further consideration for publication, the following comments and suggestions can be considered.

1. In the introduction part about literature review, the author mainly focused on the work about gripper while introducing almost works about woven meta-structure of structure. It's better to do some literature review about weaving structure or meta-structure to explain their uniqueness.
2. In figure 1, the author mentioned “z-axis” in the caption while no definitions are given in the figures. It is better to given clear definition about the axis to avoid any confusion.
3. In the study of the shape changing process of each closed-loop strip, the FEM tool is used while can the author do some comments about the theoretical prediction?
4. The author may thick change the index "A" in Fig. 2l by other indices like number to prevent any possible confusion, and as well the other related index throughout the manuscript.
5. Scale bar is missing in figure 2 and 3;
6. For the paragraph from line 222 to line 230, the author may point out which figure are these results refer to.
7. About the grasping behavior, even the author explain the entanglement mechanism for enclosing and then grasping, the information about how the gripper interact with the object especially the thin card during grasping is still missing. Some experiments can be added for this point.
8. What's the grasping limitation in terms of size, shape and fragileness of object?

Reviewer #2 (Remarks to the Author):

This article presents a weaving-inspired grasping mechanism for increasing the payload capacity without losing the softness or slenderness. The article provides inspiration on the variations of designing soft grippers. The article is well-organized, but not clearly written. Here are my comments for the author to consider for revisions.

Major Comments:

The explanation of grasping kinematics and the evaluation of grasping performance may need to be reconstructed. Fig.2 may explain clearly how to transfer the weave deformation into the grasping mechanism. Fig.3 may focus on evaluating the performance of the grasping with well-defined metrics. The reaction force is a function of rotation angle, applied torque, geometry of the objects, contacting point, and the number of strips. The data sets supplied in Fig.3 (g,h) associated with the explanations in the text are not enough to support the argument. The demonstration is impressive. However, what is the key to holding a heavy object? Or the key for enhancing the payload carrying capacity is not explicitly expressed.

Minor Comments

1, I found Fig.1 (c,d,e) confusing. What is the connection between Fig.1c and Fig.1d? How does the woven textile inspire the design of the gripper? The color code is not distinguishable. The + or - signs are not shown in Fig.1d. I found it would be difficult to imagine the transformation from Fig.1d to Fig.1e.

2, line 94, the author mentioned about the entanglement of warp and weft threads. In line 105, the author also mentioned the gripper. It may be helpful to illustrate what are warp and weft, and how they inspired the design of the gripper in Fig.1.

3, Fig.2 supposes to be the main figure to explain the grasping kinematics and the deformation of the proposed gripper. However, the description is confusing. Line 132 to line 161 may need to be rewritten.

4, Fig.2b, is the color bar representing the range of z coordinates or deflections?

5, Fig.3(a,b,c,d) shows 80% object size. And this size is written as "An object of 80% size compared to the internal gripper volume....." in line 192. What is the definition of the "internal gripper volume" here?

6, Fig.3a the author may want to supply more detailed information about how the FEA was performed either in the Methods Section or supplemental information. The color bar and the z-axis represent the same information. 7, line 222, the author talked about the contact area, is there any quantitative investigation on how contact area affects the payload carrying capacity?

Response to Reviews

Response to Reviewer 1

Dear Reviewer #1,

In the paper titled as “Grasping through dynamic weaving with entangled closed loops”, the authors provide a novel technique by using interwoven structure as gripper with high payload-to-weight ratio. The authors studied the grasping mechanism with sufficient details. Gripper with grasping mechanism initiated by the weaving of closed-loop strips is novel and promising. Before further consideration for publication, the following comments and suggestions can be considered.

Response:

We sincerely appreciate your thoughtful evaluation of our manuscript. Taking into account your comments, we have diligently revised the manuscript to further enhance its quality as follows.

1. In the introduction part about literature review, the author mainly focused on the work about gripper while introducing almost works about woven meta-structure of structure. It's better to do some literature review about weaving structure or meta-structure to explain their uniqueness.

Response:

We appreciate your invaluable advice. In accordance with your advice, we included additional literature on woven structures and meta-structures to further explain their uniqueness and strengthen the rationale.

Revised:

Line 58-64: “Weaving typically involves constructing a continuous surface, such as textiles, from discrete elements like threads^{44,45}. The mechanical entanglement of weft and warp threads yields stability, emulating the characteristics of a continuous substrate even though it is made up of individual segments^{46,47} Extensive research is being conducted on woven structures in diverse fields, including medical field, owing to their remarkable homogeneity, stability and capacity to interconnect complex structures without requiring specialized joining methods^{48,49}”

44. Alam, M. S., Majumdar, A. & Ghosh, A. Development and experimental validation of a mathematical model of shear rigidity of woven fabric structures. *J. Text. Inst.* **113**, 824-832, doi:10.1080/00405000.2021.1906489 (2022).

45. Zhang, Z.-H., Andreassen, B. J., August, D. P., Leigh, D. A. & Zhang, L. Molecular weaving. *Nat. Mater.* **21**, 275-283, doi:10.1038/s41563-021-01179-w (2022).

46. Begum, M. S. & Milašius, R. Factors of Weave Estimation and the Effect of Weave Structure on Fabric Properties: A Review. *Fibers* **10**, 74 (2022).

47. Maziz, A. *et al.* Knitting and weaving artificial muscles. *Sci. Adv.* **3**, e1600327, doi:10.1126/sciadv.1600327 (2017).

48. Liberski, A. *et al.* Weaving for heart valve tissue engineering. *Biotechnol. Adv.* **35**, 633-656, doi:https://doi.org/10.1016/j.biotechadv.2017.07.012 (2017).

49. Cox, B. N. & Flanagan, G. Handbook of analytical methods for textile composites. (1997).

2. In figure 1, the author mentioned “z-axis” in the caption while no definitions are given in the figures. It is better to given clear definition about the axis to avoid any confusion.

Response:

We apologize for any confusion that may have arisen due to the omission of important information. In order to prevent any further confusion, we added the coordinate in Fig. 1b.

Fig. 1b. Threads form the warp and weft with having position differences along the z-axis.

3. In the study of the shape changing process of each closed-loop strip, the FEM tool is used while can the author do some comments about the theoretical prediction?

Response:

We thank the Reviewer for the suggestion. In accordance with your advice, we added sentences to address the Reviewer’s comment in the Methods Section.

Revised(added):

Line 395-399: “Essentially, the deformations occurred in this study involves large displacement and strain in thin shell structures with both bending and twisting, making the structural morphing complex. Due to the difficulty of predicting them with a simple theoretical model, numerical simulations were employed to understand the complex morphing and mechanical behavior of the grippers in detail.”

4. The author may thick change the index "A" in Fig. 2l by other indices like number to prevent any possible confusion, and as well the other related index throughout the manuscript.

Response:

In accordance with your advice, we changed the thickness of the index “A”, and “B” in the Fig. 2l, Fig. 2o and Fig. 2r.

(Revised Figure)

Fig. 2 | Weaving gripper mechanism.

5. Scale bar is missing in figure 2 and 3;

Response:

In accordance with your advice, we added the scale bar in Figure 2(Fig. 2d, 2g, 2j, 2m, 2p) and Figure 3(Figure 3b, 3c, 3d, 3e and 3f).

(Revised Figures)

Fig. 2 | Weaving gripper mechanism. (Figs. 2d, 2g, 2j, 2m, and 2p)

Fig. 3 | Force prediction and deformation analysis depending on object size. (Figs. 3b-d)

Fig. 4 | Experimental analysis for weaving gripper performance. (Figs. 4a-b)

6. For the paragraph from line 222 to line 230, the author may point out which figure are these results refer to.

Response:

We thank the Reviewer for the valuable comment. We agree that we need to clarify the reference of the results to specific figures. The measured payload data, W_{max} , was obtained by conducting five measurements for each combination of rotating angle and the number of strips. We plot the W_{max} data in box-plot graph according to the rotating angle and the result, as shown in Fig. 3g. The small square at the center of each boxplot represents the average value, and the corresponding average values were mentioned in line 222 to line 230. Additionally, for reference, you can check one of the representative payload data in Supplementary Fig.7. In order to enhance comprehension, we included additional information from line 245 to line 255.

Fig. 3 | Experimental results for weaving gripper performance. g Load capacity measured depending on θ_a (red, yellow, and blue correspond to N_8 , N_{12} , and N_{16} , respectively)

Supplementary Fig. 7 | The experimental data of load capacity for N (**a** N_8 , **b** N_{12} , **c** N_{16}) using B_{90} .

Revised(added):

Line 245-255: “The performance of the gripper was found to depend on N (Fig. 4c). Because N_{16} had the most strips, and thus the thinnest width, it was relatively flexible compared with N_8 , and deformation occurred easily. However, W_{max} was relatively low owing to the weak rigidity of the strip. W_{max} measured on average at $\theta_a = 1^\circ$ was approximately 2.78 kg·f with N_{16} (The blue boxplot in Fig. 4c shows the results of five experiments with N_{16} , and the small square in the center of the boxplot shows the average value of W_{max}), and 4.04 kg·f with N_8 (The red boxplot in Fig. 4c shows the results of five experiments with N_8), approximately 1.5 times the value for N_{16} . The performance difference between samples with different N values became clearer as the rotation angle increased. It was approximately 6.8 times at $\theta_a = 20^\circ$, measured at approximately 6.15 kg·f with N_{16} and 41.60 kg·f with N_8 . The results show that a gripper with fewer strips reaches a critical point at a smaller θ_a .”

7. About the grasping behavior, even the author explain the entanglement mechanism for enclosing and then grasping, the information about how the gripper interact with the object especially the thin card during grasping is still missing. Some experiments can be added for this point.

Mechanism for grasping a thin card using the weaving gripper: a The moment that the gripper contact the edge of a thin card b A thin card is held slightly bent due to the force of the gripper trying to reach $\theta_i=0$ (the woven state).

Response:

We grateful your invaluable advice. The gripper's structural advantages include: as each strip transitions from the open to the woven state, the beveled edges of the strips enable them to pass through the narrow gap between the card and the bottom surface at an angle, thereby allowing them to securely grip even thin cards. Moreover, the gripper, which is made of flexible material, exhibits high adaptability, allowing it to adhere better to various ground shapes compared to the grasping object.

The detailed gripping mechanism for thin objects is as follows. (1) Initially, the gripper approaches to the thin object, pushing it towards the center until the strip comes into contact with the object's edge. (2) The thin beveled edges and the force of the gripper trying to reach $\theta_i=0$ (the woven state) facilitates a downward digging motion, which aids the gripper in smoothly passing underneath the card. (3) Subsequently, the gripper can effectively grip thin objects like thin cards.

In accordance with your advice, we have included an explanation from line 325 to line 327.

Revised(added):

Line 325-327: "The gripper is also capable of securely grasping thin and flat objects, such as a thin card, utilizing its thin beveled edges and exerting force trying to reach $\theta_i=0$ (the woven state)."

8. What's the grasping limitation in terms of size, shape and fragileness of object?

Response:

We appreciate your crucial question. We provide a detailed explanation regarding the gripper's capabilities with respect to object fragileness, size, and shape.

Primarily, in terms of handling fragile objects, the gripper is designed with a spatial volume closely resembling a sphere. Consequently, objects with a size equal to or less than the internal volume of the gripper can be securely gripped even if they possess low hardness or fragility because the inner volume of the gripper is maintained and no pressure is applied to the object. Additionally, when the object's volume exceeds that of the gripper, the flexible material properties of the gripper and the gap distance between the loops allows it to adapt to the original structure of most fragile objects, ensuring a gentle grip.

Regarding the shape of the object, it is primarily irrelevant to the grasping performance of gripper. However, if the object volume surpasses that of the gripper, it is necessary to have graspable geometries on the object,

such as a dumbbell-like structure, for effective grip. Any shape can be gripped if the object contains an additional composite structure that allows the weave of the gripper to form when gripped.

Considering the size of the object, it can be gripped as long as it is smaller than or equal to the internal volume of the gripper. The gripper's unique characteristic lies in its woven state, which becomes increasingly robust as θ_i returns to zero. When the volume of the object is less than or equal to 100% of the gripper's volume, a complete grip of the object is possible as the gripper reaches the woven state. As an exception, as mentioned for object geometry, larger objects can be gripped if there is a combination of structures within the geometry that collectively occupy a volume less than or equal to 100%. Even the larger objects can be gripped by friction, deformation, etc. To provide additional information, we revise the manuscript as follows.

Revised(added):

Line 316-325: “The flexibility of the material enables to grasp a wide range of objects. In our study, our primary focus was on examining spherical-shaped objects that resemble the inside of the gripper. We emphasized the variation in the number of strips based on the object's volume, and explored how this variation affects performance. As the gripper is designed with a spatial volume resembling a sphere, objects with a size equal to or less than the gripper internal volume can be securely gripped even if they possess low hardness or fragility. Regarding the shape of the object, it is primarily irrelevant to the grasping performance. However, if the volume of the object exceeds the volume of the gripper, the presence of an additional structure capable of forming a weave structure (such as a dumbbell-like structure) can be advantageous for grasping a high weight of tens of kilos or more.”

Response to Review (Reviewer 2)

Dear Reviewer #2,

This article presents a weaving-inspired grasping mechanism for increasing the payload capacity without losing the softness or slenderness. The article provides inspiration on the variations of designing soft grippers. The article is well-organized, but not clearly written. Here are my comments for the author to consider for revisions.

Response:

We deeply appreciate your thoughtful evaluation of our manuscript. We have carefully considered your comments and incorporated them into the revised manuscript as follows.

Major Comments

The explanation of grasping kinematics and the evaluation of grasping performance may need to be reconstructed.

1. Fig.2 may explain clearly how to transfer the weave deformation into the grasping mechanism.

Response:

We sincerely apologize for any confusion that may have resulted from the inadvertent omission of crucial information. For more clear understanding, we modified the explanation about how to transfer the weave deformation into the grasping mechanism.

Revised:

Line 148-168: “At $\theta_i = 0^\circ$, the θ -axis values of the inner and outer bound points match, and the gripper is in an unwoven state (Outer bound points and inner bound points are on the outer ground line and inner ground line, respectively) (Fig. 2d–f). The independent strips do not cross each other because they are sequentially arranged with original structural position differences in the x-y plane and along the z-axis. When the inner plate is rotated clockwise such that $\theta_i = -22.5^\circ$, the inner ground point shifts to -22.5° (Fig. 2g–i). When $\theta_i = -45^\circ$, the θ values of the outer bound point of the preceding strip and the inner bound point of the reference strip coincide by rotating clockwise (Fig. 2j–l). However, owing to the differences in the original position difference in the x-y plane and along the z-axis, the strips do not collide and always overlap with respect to the z-axis with a certain rule: the reference strip is always located lower than the preceding strip clockwise (Fig. 2lA) and higher than the following strip on the z-axis (Fig. 2lB), in the exact same way that warp and weft threads are interlaced in the woven structure (Fig. 1c); when $\theta_i = -90^\circ$, the inner bound point of the reference strip overtakes the outer bound point of the preceding strip along the θ -axis (Fig. 2m–o). Then, an intersection between the strips occurs (Fig. 2oA, oB), whereas the reference strip is still positioned lower along the z-axis than the preceding strip and higher than the following strip. $\theta_i = -180^\circ$ is the angle θ_s at which the inner and outer boundary points within each strip are symmetrical (Fig. 2p–r). At θ_s , the strips pass near the center point, and the number of overlapping intersections between the strips increases. Eventually, the intersection points of all the strips are firmly formed at positions lower than those of the preceding strips and higher than those of the following strips along the z-axis (Fig. 2rA, rB), forming the most stable woven state.”

2. Fig.3 may focus on evaluating the performance of the grasping with well-defined metrics. The reaction force is a function of rotation angle, applied torque, geometry of the objects, contacting point, and the number of strips.

Response:

We greatly appreciate your feedback, which has guided us in improving the structure and clarity of the figures. The Fig. 3a represents the analysis result obtained through FEA, specifically showcasing the (in-plane) reaction force when a strip is rotated by θ_i and then pushed outward to $R_{incircle}$. Due to the experimental limitation of a measurement method for the reaction force, FEA was utilized to analyze and predict its trend. Additionally, a simple experiment was conducted using three different sized balls to verify the performance, as illustrated in Figs. 3b-d).

Meanwhile, Figs. 3e-f show the set-up utilized for measuring payload and torque, and the corresponding results are presented in Figs. 3g-h. In accordance with your advice, we have made the decision to split Fig. 3 into two separate figures, Fig. 3 and Fig. 4. This adjustment aims to enhance the clarity of the figure composition. Specifically, the Fig. 3 shows the analytical results, while Fig. 4 presents the experimental results.

(Revised Figures)

Fig. 3 | Force prediction and deformation analysis depending on object size. **a** Reaction force of the gripper prediction using FEM according to $R_{incircle}$ and θ_i . The reaction force changes according to the degree of deformation of the gripper and the state of contact with the grasping target. **b** Weaving process with an 80% object size compared to the internal gripper volume. N_8 contacts the surface of the object at $\theta_i = -180^\circ$ and twists at $\theta_i = -240^\circ$. **c** The 100% object size contacts at $\theta_i = -160^\circ$ and is woven at $\theta_i = -210^\circ$. **d** The 120% object size contacts at $\theta_i = -135^\circ$ and still is not woven at $\theta_i = -190^\circ$.

Fig. 4 | Experimental analysis for weaving gripper performance. **a-b** Experimental setup for measuring load capacity and torque. **c** Load capacity measured depending on θ_a (red, yellow, and blue correspond to N_8 , N_{12} , and N_{16} , respectively). **d** Load capacity results for torque according to object size (blue, red, and yellow correspond to B_{75} , B_{90} , and B_{105} , respectively).

3. The data sets supplied in Fig.3 (g,h) associated with the explanations in the text are not enough to support the argument.

Response:

We apologize for any confusion that may have arisen due to the omission of important information.

In Fig. 3g, we present the measured payload data obtained from each five measurements, focusing on the combination of rotating angle and number of strips. We conducted tests on three different-sized balls and found that the smaller ball (B_{75}) consistently exceeded the payload limit of 100 kg·f (L_{max}) even at a rotation angle of $\theta_a = 1^\circ$, while the larger ball (B_{105}) always slipped in all grippers. To further analyze the performance based on rotation angle and number of strips, we compared the payload data of a B_{90} , which represents a similar size to the internal gripper volume. As shown in the box plot graph of Fig.3g.

In Fig. 3h, we present the results of generated torque and load capacity by different object size and number of strips N . The smaller objects B_{75} that reached the woven state exhibited lower torque (less than 2.5 N·m) and higher payload (over 100 kg·f) due to be woven at $\theta_a = 1^\circ$. While the larger objects, which failed to reach the woven state and experienced slipping, demonstrated relatively low torque and payload. To address any potential confusion, we modified and provided additional explanation for Fig. 3g-h.

Revised:

Line 241 – 276: " Balls larger or smaller than the gripper showed non-discriminatory results regardless of θ_a and N ; therefore, the difference in performance according to θ_a and N was compared using a 90 mm ball assumed to be of similar size to the internal gripper volume.

The performance of the gripper was found to depend on N (Fig. 4c). The N_{16} had the most strips, and thus the thinnest width, it was relatively flexible compared with N_8 , and deformation occurred easily. However, W_{max} was relatively low owing to the weak rigidity of the strip. W_{max} measured on average at $\theta_a = 1^\circ$ was approximately 2.78 kg·f with N_{16} (The blue boxplot in Fig. 4c shows the results of five experiments with N_{16} , and the small square in the center of the boxplot shows the average value of W_{max}), and 4.04 kg·f with N_8 (The red boxplot in Fig. 4c shows the results of five experiments with N_8), approximately 1.5 times the value for N_{16} . The performance difference between samples with different N values became clearer as the rotation angle increased. It was approximately 6.8 times at $\theta_a = 20^\circ$, measured at approximately 6.15 kg·f with N_{16} and 41.60 kg·f with N_8 . The results show that a gripper with fewer strips reaches a critical point at a smaller θ_a .

In conclusion, the control of the rotation angle θ_i between the two plates is a key factor in weaving. The load capacity increases exponentially beyond a certain critical angle, which depends on the number of strips. This tendency occurs because, as the rotation angle increases, the required force for the object to pass through the center point of the strip increases, and this force is directly related to the success of weaving. At $\theta_a = 1^\circ$ and 4.5° , the payload W_{max} was measured to be 6.5 kg·f or less; the woven structure was not reached before 4.5° . In comparison, it showed more than two times the performance at $\theta_a = 20^\circ$ and ten times the performance at 90° compared to 4.5° (Supplementary Fig. 7). When $\theta_a = 0^\circ$, the 90 mm ball was averagely wrapped only up to 88 mm, which was a height of 98% from the top of the ball, so the weaving was not formed yet. Rotating by more than 20° brought the distance between loops closer to zero sizes of $R_{incircle}$ and improved the performance by more than 10 times. Consequently, to maximize the loading capacity of the gripper, it is essential to make $R_{incircle}$ as small as possible close to the point or, equivalently, to make the interlacing of the strips as dense as possible.

The gripper exhibited a high mechanical efficiency compared to the generated torque (Fig. 4d), which was especially marked by the size of the object. The smaller objects B_{75} , which was fully gripped and reached the critical point at θ_s , W_{max} was measured over an L_{max} with a torque of less than 2.5 N·m. In contrast, the larger objects B_{105} , which failed to reach the woven state and experienced slipping, demonstrated relatively low torque and payload. A load of approximately 2% was measured with B_{105} , whereas the generated torque was similar to that of B_{75} ."

4. *The demonstration is impressive. However, what is the key to holding a heavy object? Or the key for enhancing the payload carrying capacity is not explicitly expressed.*

Response:

The weaving structure is the key to holding a heavy payload in our device. The motivation for our utilization of weaving structures comes from the well-known durability of woven fabrics and woven baskets. The weaving structure is made by interlacing many threads, or strips in our case, at a certain angle to each other. The weaving structure is known for its strong resilience to shape deformation (induced by, e.g., the payload on it) due to the cooperative efforts to sustain its shape by the constituent strips: The change of the relative position and orientation of each strip is obstructed by all the other strips that are interlocked with each other. This strong geometric resilience of the weaving structure employed in our device allows the individual strips to handle heavy payloads by distributing and supporting the weight.

Revised(added):

Line 309-313: “The weaving structure is known for its strong resilience to shape deformation (induced by, e.g., the payload on it) due to the cooperative efforts to sustain its shape by the constituent strips. This strong geometric resilience of the weaving structure employed in our gripper allows the individual strips to handle heavy payloads by distributing and supporting the weight.”

Minor Comments

1. *I found Fig.1 (c,d,e) confusing. What is the connection between Fig.1c and Fig.1d? How does the woven textile inspire the design of the gripper? The color code is not distinguishable. The + or - signs are not shown in Fig.1d. I found it would be difficult to imagine the transformation from Fig.1d to Fig.1e.*

Response:

We apologize for any confusion that may have arisen due to the insufficient information. In order to prevent any further confusion, we changed the configuration and caption of Figure 1.

Firstly, in Fig.1, our intention was to showcase the reconstruction of woven structure to our gripper. We got motivated from the durability of woven fabrics, and focused on the relative height differences between adjacent lines. The weaving structure is made by interlacing many threads, or strips in our case, at a certain angle to each other. Fig.1d and Fig.1e illustrated the open and closed state of the gripper, respectively. In the open state, where there is no entanglement, it is not possible to compare the height difference between strips along the z axis. Therefore, the (+) or (-) signs were not shown in Fig.1d. However, considering your comment about difficulty in visualization and potential confusion, we have decided to remove Fig.1d and modified the configuration of Fig. 1.

Secondly, we agree that the color code used in Fig.1 was not sufficiently distinguishable. In response to your valuable advice, we changed the color code with distinct colors.

(Revised Figure)

Fig. 1 | Principle of woven structure. **a** Woven structure in fabric. **b** Threads form the warp and weft with having position differences along the z-axis. **c** Every thread intersecting at the center with a position differences in z-axis by moving linearly. Adjacent threads are each at a relatively high (+) or low (-) position in the z-direction. **d** Weaving mechanism reconfiguration for gripper by connecting the end of each strips with making closed loop.

Revised:

Line 98-101: “The structure features threads that exist independently, are interlaced, and support each other by moving linearly to the center with having height differences (the difference in the height of each threads is described by (+) and (-) sign. For example, the white one is lower (-) than the red one in z-direction, but higher (+) than the blue one in z-direction) (Fig. 1c).”

2. line 94, the author mentioned about the entanglement of warp and weft threads. In line 105, the author also mentioned the gripper It may be helpful to illustrate what are warp and weft, and how they inspired the design of the gripper in Fig.1.

Response:

We apologize for any confusion that may have arisen due to the omission of important information. We fully agree that the explanation about the entanglement of warp and weft threads for the design of the gripper was not sufficient. Taking your advice into consideration, we included additional information in line 107-115.

Revised(added):

Line 107-115: “In Fig. 1c, the horizontal direction (parallel to the y-axis) represents the weft, indicated by the presence of red and blue strips and the vertical direction (parallel to the x-axis) corresponds to the warp, denoted by the white and yellow strips (set based on the orientation of the current figure). By connecting the ends of each strip in sequence, respectively, the lines of the strips can be changed into multiple closed loops that are intertwine with each other (Fig. 1d). The weaving mechanism then can be adapted for the gripper, and exhibits significantly enhanced compositional strength as the weaving is completed with the gripper in the closed state. The strips can be brought together linearly towards the center, resulting in a woven state, while relative rotation moves the strips to an unwoven open state.”

3. Fig.2 supposes to be the main figure to explain the grasping kinematics and the deformation of the proposed gripper. However, the description is confusing. Line 132 to line 161 may need to be rewritten.

Response:

We apologize for any confusion that may have arisen due to the omission of important information. Taking your advice into consideration, we modified the description of Fig. 2 and provided additional figure in Supplementary Fig.1.

(added Figure)

Supplementary Fig. 1 | The completeness of weaving is quantified through an imaginary inscribed circle with radius R_{incircle} , where the inner edges of the strips were tangents in weaving gripper.

Revised:

Line 148-179: “At $\theta_i = 0^\circ$, the θ -axis values of the inner and outer bound points match, and the gripper is in an unwoven state (Outer bound points and inner bound points are on the outer ground line and inner ground line, respectively) (Fig. 2d–f). The independent strips do not cross each other because they are sequentially arranged with original structural position differences in the x-y plane and along the z-axis. When the inner plate is rotated clockwise such that $\theta_i = -22.5^\circ$, the inner ground point shifts to -22.5° (Fig. 2g–i). When $\theta_i = -45^\circ$, the θ values of the outer bound point of the preceding strip and the inner bound point of the reference strip coincide by rotating clockwise (Fig. 2j–l). However, owing to the differences in the original position difference in the x-y plane and along the z-axis, the strips do not collide and always overlap with respect to the z-axis with a certain rule: the reference strip is always located lower than the preceding strip clockwise (Fig. 2lA) and higher than the following strip on the z-axis (Fig. 2lB), in the exact same way that warp and weft threads are interlaced in the woven structure (Fig. 1c); when $\theta_i = -90^\circ$, the inner bound point of the reference strip overtakes the outer bound point of the preceding strip along the θ -axis (Fig. 2m–o). Then, an intersection between the strips occurs (Fig. 2oA, oB), whereas the reference strip is still positioned lower along the z-axis than the preceding strip and higher than the following strip. $\theta_i = -180^\circ$ is the angle θ_s at which the inner and outer boundary points within each strip are symmetrical (Fig. 2p–r). At θ_s , the strips pass near the center point, and the number of overlapping intersections between the strips increases. Eventually, the intersection points of all the strips are firmly formed at positions lower than those of the preceding strips and higher than those of the following strips along the z-axis (Fig. 2rA, rB), forming the most stable woven state.

To use weaving in a gripper, it is necessary to be able to hold or release an object by using the relative rotation of the strips. The gripper performance was determined by the completeness of the weaving formed by the warp and weft strips surrounding the object. The completeness of weaving was quantified through an imaginary inscribed circle with radius R_{incircle} , where the inner edges of the strips were tangents (Fig. 2b and Supplementary Fig. 1). As the input angle increases, the interlacing of the strips becomes denser and the size of the incircle decreases. If there is no reaction force from an object, R_{incircle} approaches 0 as θ_i approaches

-180°, and it becomes the closest to the woven state. This state was set as the critical point at which the gripper reached the weave. To maximize the load capacity of the gripper, it is crucial to approach the critical point where there is a reaction force, and the gripper can be reached through a complementary relationship between θ_i and R_{incircle} .”

Line 2-4 (supplementary): “Supplementary Fig. 1 | The completeness of weaving is quantified through an imaginary inscribed circle with radius R_{incircle} , where the inner edges of the strips were tangents in weaving gripper.”

4. Fig.2b, is the color bar representing the range of z coordinates or deflections?

Response:

We apologize for any confusion that may have arisen as a result of the missing important information. In Fig. 2b, the color bar represents the range of z coordinates, not deflections. To avoid any further misunderstanding, we included additional explanatory sentences in the caption of Fig. 2b.

Revised(added):

Line 141-142: “b Schematic of one strip forming a loop on the x-y plane (color bar representing the range of z coordinates)”

5. Fig.3(a,b,c,d) shows 80%, object size. And this size is written as "An object of 80% size compared to the internal gripper volume....." in line 192. What is the definition of the "internal gripper volume" here?

Response:

We sincerely apologize for any confusion caused by the oversight in failing to include important details. We assumed that the gripper has an ideal spherical shape. To determine the diameter of the gripper, we measured the distance from the top to the bottom of the closed gripper using a pointer laser sensor (IL-600, Keyence), which resulted in an approximately 90 mm. We used this measurement as a reference for the diameter of the gripper, assuming that it is comparable to the internal volume of the gripper. In our previous explanation, we indirectly referenced the internal volume of the gripper in line 194-195. However, we agree that this explanation was not sufficient. Taking your advice into account, we included additional information in supplementary Fig. 2 to provide more detailed understanding of the internal volume of the gripper.

(Added Figure)

Supplementary Fig. 2 | The internal volume of the gripper: We assumed that the gripper has an ideal spherical shape. To determine the diameter of the gripper, we measured the distance from the top to the bottom of the closed gripper using a pointer laser sensor (IL-600, Keyence), which resulted in an approximately 90 mm. The diameter of the 90 mm ball is 100% object size compared to the internal gripper volume, assuming that the diameter of the 90 mm sphere is similar to the internal volume of the gripper.

Revised(added):

Line 199-200: “Assuming that the strips form an ideal sphere at θ_s ($\theta_i = -180^\circ$), the sphere diameter is approximately 90 mm (Supplementary Fig. 2).”

Line 8-13 (supplementary): “Supplementary Fig. 2 | The internal volume of the gripper: We assumed that the gripper has an ideal spherical shape. To determine the diameter of the gripper, we measured the distance from the top to the bottom of the closed gripper using a pointer laser sensor (IL-600, Keyence), which resulted in an approximately 90 mm. The diameter of the 90 mm ball is 100% object size compared to the internal gripper volume, assuming that the diameter of the 90 mm sphere is similar to the internal volume of the gripper.”

6. Fig. 3a the author may want to supply more detailed information about how the FEA was performed either in the Methods Section or supplemental information. The color bar and the z-axis represent the same information.

Response:

We apologize for any confusion that may have arisen due to the omission of important information. In order to prevent any further confusion, we added sentences to deal with the Reviewer’s comment as followings.

Also, as you mentioned, it is true that the color bar and the z-axis convey the same information. While this may seem redundant, we chose to use both the z-axis and the color bar simultaneously to provide an intuitive understanding. By presenting the information in this way, we aimed to ensure clarity and facilitate a comprehensive interpretation of the data.

Revised(added):

Line 405-410: “In Fig. 3, we performed numerical experiments for a single strip to get some insights during grasping process. First, we bent a straight strip to form a purely bent shape ($\theta_i = -180^\circ$). Then, we measured the in-plane reaction forces while rotating and pushing it outward from the center point. Specifically, we rotated the strip to θ_i and then applied the in-plane displacement without constraining its vertical movement, pushing the middle part outward so that the center inner face of the strip is on $R_{incircle}$.”

7. line 222, the author talked about the contact area, is there any quantitative investigation on how contact area affects the payload carrying capacity?

Response:

Fig. 3g presents the effect of the number of strips. We assumed and illustrated that the contact area would vary with the number of strips, but as you pointed out, it was difficult to derive a quantitative relationship and confusing to explain, so we removed and simplified it. All design parameters of the grippers (N_8 , N_{12} , and N_{16}) were kept the same, and only the outer plate was varied by configuring the polygons within the same outer circle. By adjusting the width of the strip relative to a polygon with N sides within the same outer circle, increasing N brings the total area closer to the surface of a sphere. Additionally, as N increases, the width of each strip becomes thinner, resulting in a relatively higher flexibility. Consequently, for the same 90 mm ball, the contact area of N_{16} is relatively larger than that of N_8 . However, as mentioned in lines 222-230, repeated experiments have shown that increasing N leads to a decrease in stiffness due to the width of the thinner strips,

resulting in a reduction in the measured average load availability. To avoid any further confusing, we modified the line 222 and, we added further details in supplementary Fig.3 to enhance comprehension.

Revised(added):

Line 245: “The performance of the gripper was found to depend on N (Fig. 4c).”

Line 21-25 (supplementary): “Supplementary Fig. 3 | The weaving grippers classified by N (considered in Fig. 3 of the main text): N_8, N_{12}, N_{16} . Each strip is designed in a linear shape with plates at both ends. The outer plate is a polygon with a 70mm diameter, and the inner plate is circular with a 30mm diameter. The total strip length to outer plate radius ratio is 10:1. All the design parameters of the grippers (N_8, N_{12} , and N_{16}) were kept the same, and only changed the outer plate by configuring the polygon within the same outer circle. By adjusting the width of the strip based on the polygon with N sides in the same outer circle, increasing N brings the total area closer to the surface of a sphere.”

REVIEWERS' COMMENTS

Reviewer #1 (Remarks to the Author):

The authors have addressed all my comments and now this work can be published.

Reviewer #2 (Remarks to the Author):

Dear Authors,

Thank you for addressing my comments. I appreciate your effort to improve the manuscript.